# Phylogenetic Analysis of the *PR-4* Gene Family in Euphorbiaceae and Its Expression Profiles in Tung Tree (*Vernicia fordii*)

**DOI:** 10.3390/plants12173154

**Published:** 2023-09-01

**Authors:** Chengbo Yang, Yaqi Yi, Jiabei Wang, Liu Ge, Lin Zhang, Meilan Liu

**Affiliations:** Key Laboratory of Cultivation and Protection for Non-Wood Forest Trees, Ministry of Education, Central South University of Forestry and Technology, Changsha 410001, China; ychengbo2022@163.com (C.Y.);

**Keywords:** tung tree, *PR-4*, evolution, inflorescence bud, expression profiles

## Abstract

Pathogenesis-related protein-4 (*PR-4*) is generally believed to be involved in physiological processes. However, a comprehensive investigation of this protein in tung tree (*Vernicia fordii*) has yet to be conducted. In this study, we identified 30 *PR-4* genes in the genomes of Euphorbiaceae species and investigated their domain organization, evolution, promoter cis-elements, expression profiles, and expression profiles in the tung tree. Sequence and structural analyses indicated that *VF16136* and *VF16135* in the tung tree could be classified as belonging to Class II and I, respectively. Phylogenetic and Ka/Ks analyses revealed that *Hevea brasiliensis* exhibited a significantly expanded number of *PR-4* genes. Additionally, the analysis of promoter cis-elements suggested that two *VfPR-4* genes may play a role in the response to hormones and biotic and abiotic stress of tung trees. Furthermore, the expression patterns of *VfPR-4* genes and their responses to 6-BA, salicylic acid, and silver nitrate in inflorescence buds of tung trees were evaluated using qRT-PCR. Notably, the expression of two *VfPR-4* genes was found to be particularly high in leaves and early stages of tung seeds. These results suggest that *VF16136* and *VF16135* may have significant roles in the development of leaves and seeds in tung trees. Furthermore, these genes were found to be responsive to 6-BA, salicylic acid, and silver nitrate in the development of inflorescence buds. This research provides valuable insights for future investigation into the functions of *PR-4* genes in tung trees.

## 1. Introduction

The tung tree (*Vernicia fordii*), a member of the Euphorbiaceae family, has been cultivated in China for thousands of years and is also grown in other countries such as Argentina, Paraguay, and Brazil [1]. It is an important oil-producing woody plant that possesses tung seed kernels with an oil content of up to 70% [1]. Tung oil, renowned for its exceptional drying properties, holds a crucial position in industrial production [1,2]. Notably, it serves as a fundamental ingredient in the creation of environmentally friendly coatings, polyurethane, thermosetting polymer, and biodiesel [2]. The chemical industry recognizes tung oil’s market competitiveness due to its eco-friendliness [3]. Nevertheless, the industry’s progress in the tung tree sector remains limited due to its low yield. The tung tree, being a monoecious plant species, exhibits a diminished proportion of female to male flowers (approximately 1:27) and functional abnormalities in female flowers, thereby leading to a reduced fruit yield [4,5]. Previous research has indicated that the development of female flowers is typically accompanied by stamen abortion, as failure to do so results in the production of infertile ovules. Salicylic acid (SA) plays a crucial role in the development of female flowers, particularly in the induction of stamen abortion [6,7].

*Pathogenesis-related* (PR) genes contribute positively to the SA signaling pathway, and their involvement extends to plant growth and disease resistance [8,9]. PR proteins have been categorized into 17 families based on various factors such as their function, serological relationship, amino acid sequence, molecular weight, and other properties. These families are denoted as *PR-1* to *PR-17* [10,11]. The *PR-4* gene family consists of proteins that possess a Barwin-like domain in the C-terminus, and their main functionality is derived from this domain [12]. Within the *PR-4* gene family, there are two distinct categories based on the presence or absence of a chitin-binding domain. Class I members contain a chitin-binding domain, whereas class II members lack this particular component [13,14]. Two categories can be distinguished based on enzymatic analyses. Class I *PR-4s* are classified as chitinases as they can bind to chitin and exhibit chitinase activity, while class II *PR-4s* demonstrate RNase and DNase activity [15].

The *PR-4* gene family was initially discovered and documented in potatoes and, since then, it has been purified, classified, and studied in various organisms, including pepper (*Capsicum chinense*), wheat (*Triticum aestivum*), sugarcane (*Saccharum officinarum*), cacao (*Theobroma cacao*) and cassava (*Manihot esculenta*) [16,17,18,19,20]. In apple (*Malus pumila*), *MdPR-4* plays a significant role in the development of floral organs, and its expression is induced by SA treatment in leaves [21]. In maize (*Zea mays*), *ZmPR-4* exhibits high expression levels in germinated seeds, particularly in the presence of silver nitrate (SN) [22]. The *PR-4* gene in wheat is involved in regulating leaf development under nitrogen stress and also contributes to root development following treatment with SA [23,24]. Similarly, in *Ficus awkeotsang*, the *FaPR-4* gene in the pericarp of achenes collaborates with other components to safeguard the nutritious seeds from abiotic stresses during summer fruit ripening [25].

Although the function of the *PR-4* gene family has been reported in many plants, a thorough and systematic investigation of this gene family in the developmental processes of the tung tree is still lacking. In previous studies, it was observed that the expression of *VfPR-4b* is significantly elevated during the 20 days before flowering of female flowers in the tung tree. This heightened expression is closely associated with the pathways involved in the synthesis and signaling of salicylic acid (SA) [7]. However, our understanding of *VfPR-4* and its potential roles in the developmental processes of the tung tree remains limited. Fortunately, the advent of high-throughput sequencing technology has facilitated the determination and publication of genome data for economically significant tree species in the Euphorbiaceae family, such as *Vernicia fordii*, *Hevea brasiliensis*, *Ricinus communis*, *Manihot esculenta*, and *Jatropha curcas* [26,27,28,29,30]. The tung tree exhibits the most significant genetic affinity with four species from the Euphorbiaceae family, as determined through evolutionary analysis of a single-gene family. Consequently, this research comprehensively examined gene structures, conserved motifs, phylogenetic evolution, gene duplications, and cis-acting elements in five Euphorbiaceae species. Additionally, the study investigated the expression patterns of these genes in various tissues, seed development stages, and inflorescence buds in response to 6-benzylaminoadenine (6-BA), SA, and SN. Our research aims to provide a comprehensive understanding of the *VfPR-4* gene family and explore its functions during seed development and inflorescence bud development, thereby providing a theoretical basis for further research.

## 2. Results

### 2.1. Identification and Characterization Analysis of the PR-4 Family

By using the Barwin domain (PF00967), we finally identified 30 *PR-4* genes in the genomes of *Vernicia fordii* (two members), *Hevea brasiliensis* (eleven members), *Ricinus communis* (three members), *Manihot esculenta* (five members), *Jatropha curcas* (four members), *Arabidopsis thaliana* (one member), and *Populus trichocarpa* (four members). Only two *PR-4* genes (*VF16136* and *VF16135*) were found in the tung tree genome. The sequence alignment analysis showed that 30 *PR-4* genes in seven species all had a Barwin domain consisting of 119 amino acids and a signal peptide consisting of 23 to 26 amino acids in the N-terminus. The 30 *PR-4* genes could be classified into two categories according to whether they had a chitin-binding domain [13]. Class I (Ia and Ib) *PR-4* genes had a chitin-binding domain, hinge domain, and Barwin domain, while Class II (IIa and IIb) *PR-4* genes only had the Barwin domain. All the *PR-4* proteins in Class Ia had a vacuolar signal except for *Potri.005G054000*, *Potri.013G041700*, and *Potri.013G041900*. The Class Ib and Class II *PR-4* proteins had no vacuolar signal (Figure 1). According to the protein sequence alignment and hidden code model, we found six conserved cysteine residues in the *PR-4* gene family, including Cys-52, Cys-73, Cys-84, Cys-87, Cys-107, and Cys-143 of *VF16136*, and Cys-99, Cys-120, Cys-131, Cys-134, Cys-154, and Cys-190 of *VF16135*. 

### 2.2. Evolutionary Relationship Analysis of the PR-4 Family

In order to study the evolutionary relationship of the *PR-4* family, we performed the phylogenetic tree of 30 PR-4 proteins in seven species. The results indicated that all PR-4 proteins were divided into four main classes (Class Ia, Class Ib, Class IIa, and Class IIb) according to separate branches in the phylogenetic tree (Figure 2a). Class Ia and Class Ib had thirteen and five members, respectively, while Class IIa and Class IIb had eight and four members, respectively (Figure 2b). The two members of *PR-4* in the tung tree belonged to Class Ia (*VF16135*) and Class IIa (*VF16136*). Interestingly, according to the classification statistics of evolutionary trees, we found that six species had Class I members, except for *Jatropha curcas*, and four members of *Jatropha curcas* all belonged to Class IIb (Figure 2c). In addition, *Hevea brasiliensis* had 11 members, which was the highest number among the seven species. Phylogenetic analysis showed that these adjacent genes in each cluster had a close relative relationship. These results suggest that these paralogues may have resulted from gene duplications in *Hevea brasiliensis*.

### 2.3. Conserved Motifs and Gene Structure in the PR-4 Family

The 30 protein sequences of the *PR-4* family were analyzed using MEME. Five conserved motifs were identified and varied from residues 22 to 31. Motifs 1–4 represented the Barwin domain, and motif 5 represented the chitin-binding domain. Each PR-4 protein contained different numbers of conserved motifs, ranging from motifs 1 to 5. For example, the 16 members in Class I all had 5 motifs, except for *XP_021650928* of *Hevea brasiliensis* and *Manes.09G152300* of *Manihot esculenta*. Interestingly, motif 5 only belonged to Class I, which consisted of the sequence alignment (Figure 3a). Additionally, the gene structure was highly conserved in the *PR-4* family. We found that 27 (90%) *PR-4* genes had only one intron and only *Potri.005G054000* of *Populus trichocarpa*, *XP_021656801* and *XP_021656802* of *Hevea brasiliensis* had two introns (Figure 3b). In addition, *VF16135* of tung tree had the longest intron of all members of the *PR-4* family.

### 2.4. Orthologous and Paralogous Genes of the PR-4 Gene Family

The homologous gene pairs contained 24 (80%) *PR-4* genes in 6 species. The alignment result of orthologous and paralogous genes showed that there were 12 pairs of paralogous genes (red line) and 50 pairs of orthologous genes (black line) in 6 species (Figure 4a–d). Among the paralogous genes, seven pairs were found in *Hevea brasiliensis*, which was greater than the other species. This result indicated that the eight *PR-4* genes of *Hevea brasiliensis* might be produced by gene duplication. Among the orthologous genes, eight pairs were observed between *Hevea brasiliensis* and *Manihot esculenta*, which was more than other groups (Figure 4a). In the tung tree, a relationship was not observed between *VF16136* and *VF16135*. However, they all had more paralogous genes with *Hevea brasiliensis* (four pairs) than the other species. The results showed that *PR-4* members of the tung tree had the closest genetic relationship with *Hevea brasiliensis*, which was consistent with the evolutionary phylogenetic tree analysis. In addition, the Ka, Ks, and Ka/Ks of these orthologous gene pairs were calculated using Ka/Ks calculator software. Our results showed that the Ka/Ks values of all *PR-4* gene family members were less than one (Figure 4e–g). This result indicated that the *PR-4* gene family was strongly purified, selected, and functioned conservatively in the evolutionary process.

### 2.5. Analysis of Cis-Acting Elements in PR-4 Promoters

Cis-regulatory elements play crucial roles in regulating plant development. Based on the PlantCARE database, we found a total of 23 cis-acting elements in 30 *PR-4* genes, including ARE, GT1-motif, TCT-motif, I-box, GATA-motif, MBS, LTR, TC-rich repeats, WUN-motif, ABRE, TGACG-motif, CGTCA-motif, TGA-element, TCA-element, AuxRR-core, P-box, O2-site, GARE-motif, TATC-box, CAT-box, GCN4-motif, circadian, and RY-element (Figure 5a). These 23 cis-acting elements were identified in the following three categories of cis-elements: plant growth and development, biotic and abiotic stress responses, and phytohormone responses in the promoter regions (Figure 5b). In the growth and development category, cis-acting elements were found extensively in the promoter regions, including the CAT-box, GCN4-motif, circadian, and RY-element regions. The most abundant cis-acting element during plant growth and development was CAT-box, accounting for 57.1%, which was related to meristem (Figure 5c). In the phytohormone-responsive category, the ABRE cis-acting element associated with abscisic acid (ABA) accounted for 20.5% and appeared 31 times (Figure 5d). In the biotic and abiotic stress response category, the ARE cis-acting element, which was essential for the anaerobic induction, accounted for the largest portion in this category at 27.8% (Figure 5e). In the tung tree, *VF16136* exhibited five cis-acting elements associated with plant hormone response, while *VF16135* displayed five cis-regulatory elements in response to biotic and abiotic stress (Figure 5b). This finding suggests that *VF16136* may play a role in hormone responses in the tung tree, while *VF16135* may be involved in biotic and abiotic stress.

### 2.6. Expression Profile Analysis of PR-4 Genes in Tung Tree

To further understand the function of the *PR-4* gene family in tung trees, we used roots, leaves, mature flowers (including male, female, and bisexual flowers), and tung seeds (including five stages of seeds) as samples for qRT-PCR analysis. The results showed that *VF16136* was significantly upregulated in leaves and bisexual flowers, and significantly downregulated in female and male flowers. *VF16135* was significantly upregulated in leaves, and significantly downregulated in female, male, and bisexual flowers (Figure 6a,b). In tung seeds, *VF16136* and *VF16135* showed the highest expression in the early stage (15 WAF) of tung seeds and were significantly downregulated in 20 WAF, 25WAF, and 30 WAF (Figure 6c,d). These results suggested that *VF16136* and *VF16135* might play an important role in leaf and seed development in tung trees.

In tung trees, the application of 6-BA has been observed to induce the transformation of male inflorescences into female inflorescences [31]. In order to investigate the impact of 6-BA on the *PR-4* gene family in tung tree inflorescence buds, we conducted an analysis. Our findings revealed a significant upregulation of *VF16136* expression after 12 h of 6-BA treatment (12 HA6-BA), while *VF16135* expression was significantly upregulated at 6 HA6-BA (Figure 6e,f). Furthermore, we observed significant repression of *VF16136* and *VF16135* expression at 5 days and 30 days after 6-BA treatment (5 DA6-BA and 30 DA6-BA), respectively, in different developmental stages of inflorescence buds. However, at 90 DA6-BA, both *VF16136* and *VF16135* expression were significantly induced (Figure 6g,h). Furthermore, the expression of *VF16136* was notably upregulated at a concentration of 6 mmol/L of SA in the inflorescence bud. Conversely, the expression of *VF16135* exhibited a positive correlation with increasing SA concentration (Figure 6i,j). Moreover, our findings revealed that both *VF16136* and *VF16135* were downregulated in the inflorescence bud at 55 days after SN treatment (DASN) (Figure 6k,l). These observations suggest that *VF16136* and *VF16135* may play a role in the developmental processes of the inflorescence bud following treatment with 6-BA, SA, and SN in the tung tree.

Additionally, we generated fusion constructs of *VF16136-GFP* and *VF16135-GFP* and examined their expression patterns in tobacco (*Nicotiana benthamiana*) leaves (Figure 6m,n). Our findings revealed that *VF16136-GFP* was specifically localized in the chloroplast, whereas *VF16135-GFP* exhibited co-localization in both the chloroplast and nucleus. These results suggest that *VfPRs* play a role in the leaves of tung trees, consistent with their gene expression. Furthermore, the observation of *VF16135-GFP* localization in the nucleus implies that *VF16135* may possess additional crucial functions in the tung tree.

## 3. Discussion

### 3.1. Evolution of PR-4 Gene Family Members in Euphorbiaceae

A previous study showed that *PR-4* genes mainly contained chitin binding, hinge, and Barwin domains [15]. Based on the conserved domain structure and phylogenetic tree of 30 *PR-4* genes in *Arabidopsis thaliana*, *Populus trichocarpa*, *Vernicia fordii*, *Jatropha curcas*, *Ricinus communis*, *Hevea brasiliensis*, and *Manihot esculenta*, PR-4 proteins were classified into Class I and Class II [13,15]. The present research suggested that Class II would appear when Class I lost the chitin-binding domain. Interestingly, *Arabidopsis thaliana* and *Populus trichocarpa* had only Class I members and *Jatropha curcas* had only Class II members in the *PR-4* gene family. This result suggested that Class I members of *Jatropha curcas* ultimately evolved into Class II members in the *PR-4* gene family. In the tung tree, there were two members of the *PR-4* gene family. *VF16135* belonged to Class Ia with the chitin-binding domain in the *PR-4* gene family. However, *VF16136* lacked the chitin-binding domain, which belonged to Class IIa in the *PR-4* gene family. In *Hevea brasiliensis*, there were 11 members in Class I and Class II of the *PR-4* gene family, with the most members being found in the *PR-4* gene family. Seven pairs of paralogous genes were also found in *Hevea brasiliensis*. Gene duplications have been considered a main force that primarily leads to the expansion of the *PR-4* gene family. The expansion in the number of genes might indicate that their functions are more complex. Natural pressure and evolutionary forces can affect duplicate genes and relevant proteins, and their mechanism is usually investigated through natural selection analysis [32]. In tung tree, all *PR-4* genes in the tung tree were found to have undergone purification during the evolutionary process. Moreover, the *PR* gene families of different species have similarly evolved through purification selection [33]. This result suggested that the conserved domain was critical and played essential functions in the antifungal activity of *PR-4*.

### 3.2. Characteristics of PR-4 Gene Family Members in Euphorbiaceae

The number of introns and exons could reflect the characteristics of genes and different combinations of exons and introns were responsible for gene functions. The gene with rapidly changing expression levels in response to stress contain significantly lower intron densities in plant [34]. PR-4 proteins directly respond to various stimuli, making them need to respond quickly. Among the 30 *PR-4* genes, 27 members contained only one intron, and three had two introns. Similar results were found in other *PR* gene families [33]. In addition, the tandem motif (motifs 1–4) was identified in 30 *PR-4* genes, which were presumed to belong to the Barwin domain based on the results from the InterProScan webserver. Motif 5 was identified in only Class I *PR-4* genes, which were presumed to be the chitin-binding domain. Motif 5 was a critical conserved motif with a potentially important function in controlling plant pathogenesis. In addition to the essential cis-acting elements, the CAT-box, ABRE, and ARE frequently appeared in the promoter region of *PR-4* genes in seven species. CAT-box is related to meristem expression, verifying that this gene may play a potential role in the floral meristem [35]. ABRE is involved in abscisic acid responsiveness [36]. ARE is essential for anaerobic induction [37]. These results indicated that the *PR-4* gene family might be crucial in regulating the stress response and plant development.

### 3.3. VfPR-4 May Participate in the Development of Leaves and Seeds in Tung Tree

In plants, various members of the *PR* family play significant roles in development [15,38,39]. For example, *DcPR-3* and *DcPR-4* type chitinases are crucial for embryogenesis to proceed beyond the globular stage in carrots (*Daucus carota*) [40]. Wheat exhibits a high accumulation of *TaPR-4* in the endosperm of its seeds [41]. Similarly, the *CaPR-5* gene in *Capsicum annuum* demonstrates high expression levels in young leaves, with a slight increase in *CaPR-5* mRNA during fruit ripening [42]. In *Cucumis melo*, certain *CmPR-5* genes exhibit high expression in young leaves, suggesting their potential influence on early leaf development [33]. In *Halostachys caspica*, the HcPR-10 protein has been found to play a role in the regulation of flowering time and the promotion of branch growth [39]. This protein interacts with various biological ligands, such as phytohormones, proteins, fatty acids, amino acids, phenolics, and alkaloids [43]. The accumulation of PR proteins in grapevine (*Vitis vinifera*) is observed during the development of berries [44]. Furthermore, the synthesis of PR proteins may occur in either the apex or lower leaves of healthy tobacco plants [45,46]. The *PR-4* family was found to be expressed in various parts of the tung tree, including the roots, leaves, and mature flowers (including male, female, and bisexual flowers). Notably, the highest expression of the *PR-4* family was observed in the leaves of two tung tree members. Additionally, significant expression of *VfPR-4* genes was detected in tung seeds at an early stage (15 WAF) [47]. Based on these findings, it could be inferred that the *PR-4* genes of the tung tree likely played a crucial role in the development of tung leaves and seeds.

### 3.4. VfPR-4 Proteins Function in Response to 6-BA and SA

Numerous studies have demonstrated the association between plant *PR* genes and various hormone responses. This correlation has been observed in grapevine, wheat, tomato (*Lycopersicon esculentum*), and tobacco plants, where the induction of *PR-4* genes occurs through the application of exogenous MeJA and SA treatment [23,47,48,49]. Similarly, the expression of *ZmPR-4* genes is stimulated by ABA or MeJA in the leaves of maize [22]. Furthermore, *SsPR-10* expression in *Solanum surattense* is induced by SA, MeJA, GA3, and ABA [50]. Birch (*Betula verrucosa*) plants exhibit the capability of BvPR-10 protein to bind cytokinins and other hydrophobic ligands, facilitating their translocation across membranes [51]. The tung tree holds significant economic value as a woody oil plant. The inadequate ratio of female flowers in tung orchards has been identified as a contributing factor to the low fruit yield [6]. Further, 6-BA, a crucial signaling molecule, plays a role in the initiation of female flower development in the inflorescence buds of tung tree. Our study demonstrated that the expression levels of *VF16136* and *VF16135* were significantly increased at 90 DA6-BA and 120 DA6-BA, while significantly decreased at 5 DA6-BA and 30 DA6-BA. Additionally, cis-acting element analysis revealed that *VF16135* possessed a TCA-element of SA. The expression level of *VF16135* exhibited a significant increase in inflorescence buds subjected to varying concentrations of SA, while *VF16136* showed significant upregulation only at a concentration of 6 mmol/L under SA-induced stress in tung tree inflorescence buds. This finding implies that *VfPR-4* genes play a crucial role in the development of inflorescence buds following treatment with 6-BA and SA. Previous studies have extensively documented the involvement of the *PR* gene family in hormonal responses. However, the precise mechanism of action in distinct species and specific individuals remains uncertain. More information about the *VfPR*s reported here is needed to understand better the role played by these molecules in tung trees.

## 4. Materials and Methods

### 4.1. Plant Materials

The stems, leaves, roots, and flowers (male, female, and bisexual) were collected from a 6-year-old tung tree ‘Putaotong’ grown at the Central South University of Forestry and Technology (Qingping Town, Yongshun County, China) under natural conditions. The seeds were collected from an 8-year-old tung tree ‘Putaotong’, including seeds at 10 WAF, 15 WAF, 20 WAF, 25 WAF, and 30 WAF. The inflorescence buds of androecious tung trees were treated with 640 mg/L 6-BA solution containing 0.05% Tween-20 in June 2022. Inflorescence buds were collected at 0, 3, 12, 24, 36, and 48 HA6-BA, and 5, 20, 30, 60, 120, and 150 DA6-BA. The inflorescence buds of bisexuality were treated with 1000 mg/L SN solution in July 2022 and were collected at 0, 10, and 55 DASN. Meanwhile, the inflorescence buds of bisexuality were treated with 0, 2, 4, 6, 8, and 10 mmol/L SA solution in March 2023 and were collected 7 days after SA was treated. 

### 4.2. Identification and Sequence Analysis of PR-4 Family

The genome of *Arabidopsis thaliana* was downloaded from the Arabidopsis Information Resource (TAIR10) database (http://www.arabidopsis.org/, accessed on 11 August 2020), and the genomes of *Populus trichocarpa* and *Ricinus communis* were downloaded from the Phytozome database (https://phytozome-next.jgi.doe.gov/, accessed on 11 August 2020). The *Hevea brasiliensis* genome was downloaded from the National Center for Biotechnology Information (NCBI) database (https://www.ncbi.nlm.nih.gov/, accessed on 11 August 2020), and the *Manihot esculenta* genome was downloaded from the Ensembl database (http://plants.ensembl.org/Manihot_esculenta/Info/Index, accessed on 11 August 2020). The *Jatropha curcas* genome was downloaded from the *Jatropha curcas* genome database (https://www.kazusa.or.jp/jatropha/, accessed on 15 August 2020). The genome of *Vernicia fordii* was obtained from the National Genomics Data Center (OR105736 and OR105737). The candidate sequences were further screened by searching for the Barwin domain (PF00967) and using HMMER [52]. All putative *PR-4* genes were further verified for the presence of the Barwin domain by submitting them to InterProScan [53], Pfam [54], and SMART databases [55].

### 4.3. Phylogenetic Analysis and Sequence Characterization

A total of 30 *PR-4* genes in seven species were included in the phylogenetic analysis. Multiple sequence alignments of all PR-4 proteins were carried out with the MUSCLE (https://www.ebi.ac.uk/Tools/msa/muscle/, accessed on 29 October 2020). Subsequently, the Neighbor-joining (NJ) tree was generated by MEGA software (v7.0.21) with bootstrap analysis (1000 replicates). Then, we analyzed the conserved domain of the PR-4 family protein by DNAMAN software (v6.0) and made analysis diagrams of the structure domain. Gene structure diagrams were produced by TBtools [56]. The motif logos of the *PR-4* family were generated using the online MEME program (http://meme.nbcr.net/meme/cgi-bin/meme.cgi, accessed on 10 November 2020) [57].

### 4.4. Identification of Paralogous and Orthologous Genes

Orthologues and paralogues were identified by using online OrthoVenn2 (https://orthovenn2.bioinfotoolkits.net/home, accessed on 21 November 2020) [58]. Ka and Ks values were calculated by TBtools [56]. Ka/Ks represents the ratio between the non-synonymous substitution rate (Ka) and synonymous substitution rate (Ks) of two protein-coding genes, and the radio is used to determine whether there are selection pressures acting on protein-coding genes. 

### 4.5. Analysis of Cis-Acting Elements in PR-4 Genes

To determine the cis-acting elements, we first obtained the promoter sequences (the 2000 bp of genomic DNA sequence upstream of the initiation code) by TBtools software (v1.108). To determine the cis-acting elements, we obtained the promoter sequences by TBtools, which is the 2000 bp of genomic DNA sequence upstream of the initiation code (ATG). Then, these promoter sequences were submitted to the PlantCARE website (http://bioinformatics.psb.ugent.be/webtools/plantcare/html/, accessed on 24 November 2020) to identify the presence of different cis-acting elements [59]. 

### 4.6. RNA Extraction and Real-Time Quantitative PCR Analysis

The total RNA of the different tissues was extracted using the RNAprep Pure Plant Kit SK1322 (Sangon Biotech, Shanghai, China). Genomic DNA was removed and the first-strand cDNA was synthesized by using HiScript II Q RT SuperMix for qPCR (gDNA wiper) (Vazyme, Nanjing, China). Real-time quantitative PCR (RT-qPCR) was performed using the SYBR Premix ExTaq II (Takara, Japan) on a CFX96 instrument (Bio-Rad Laboratories, Hercules, CA, USA) according to the manufacturer’s instructions. The qRT-PCR primers for *VfPR-4* were designed by Primer Premier 5 (Appendix A). Tung tree elongation factor 1-α (*EF1α*) was used as the reference gene [60]. The relative expression levels were calculated by using the 2^−ΔΔCT^ method [61].

### 4.7. Subcellular Location Analysis

The InFusion cloning system was used to construct plant overexpression vectors pCAMBIA1300-GFP:*VF16136* and pCAMBIA1300-GFP:*VF16135*. The restriction sites *EcoRI* and *SalI* were used to line the pCAMBIA1300-GFP vector. Two recombinant plasmids were transformed into *Agrobacterium tumefaciens GV3101* using the liquid nitrogen rapid freezing method. *Agrobacterium tumefaciens* cultures were diluted in infiltration buffer until optical density (OD_600_) = 1 and then infiltrated by syringe on the abaxial *Nicotiana benthamiana* leaf surface. The leaves were then sampled for fluorescence observation. DAPI staining solution was used to mark the nucleus in the leaves. The fluorescence was observed 2–3 days later by an LSM 510 confocal laser scanning microscope (Carl Zeiss AG, Oberkochen, Germany) with the following parameters: fluorescence imaging of GFP excitation at 488 nm and scanning at 495–545 nm; chloroplast excitation at 545 nm and scanning at 585 nm; and DAPI excitation at 360–400 nm and scanning at 430–550 nm.

### 4.8. Statistical Analysis

The expression pattern analysis data were analyzed by Excel 2016 and SPSS 17.0 (SPSS Inc., Chicago, IL, USA) software. The data presented were indicative of the mean and standard deviation (SD) of three separate biological replicates. The results were statistically analyzed by one-way analysis of variance (ANOVA) followed by the Games–Howell test. In all figures presented, the error bars indicated the SD.

## 5. Conclusions

In conclusion, we comprehensively analyzed the domain organization, phylogenetic relationships, promoter cis-elements, expression profiles of the *PR-4* gene family, and interactions of PR-4 protein in the tung tree. The sequence and motif analysis indicated that the domain structures of all genes were conserved. The *PR-4* gene had a greater degree of expansion in the genome of *Hevea brasiliensi* than in other plant species. Furthermore, the cis-activated element analysis implied that *PR-4* could participate in plant growth and development, biotic and abiotic stress responses, and phytohormone responses in the promoter regions. The differential expression patterns of *VfPR-4* genes in various tissues and inflorescence buds observed after the 6-BA, SA, and SN treatment suggested that *VfPR-4* genes might play an important role in the development and reproductive physiology of the tung tree.

## Figures and Tables

**Figure 1 plants-12-03154-f001:**
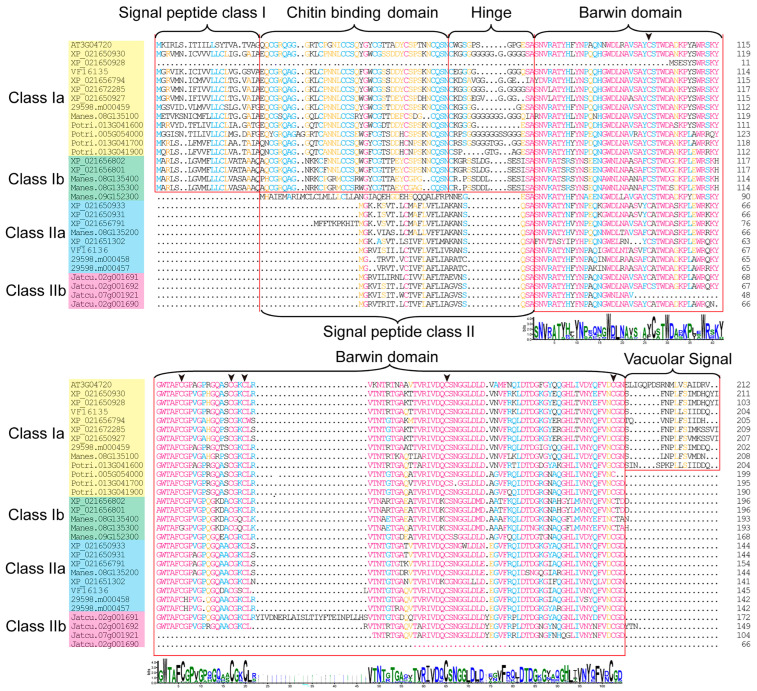
Sequence classification analysis of *PR-4* protein in seven species. Different classes are marked in different colors The arrows marked six conserved cysteine residues in the *PR-4* gene family.

**Figure 2 plants-12-03154-f002:**
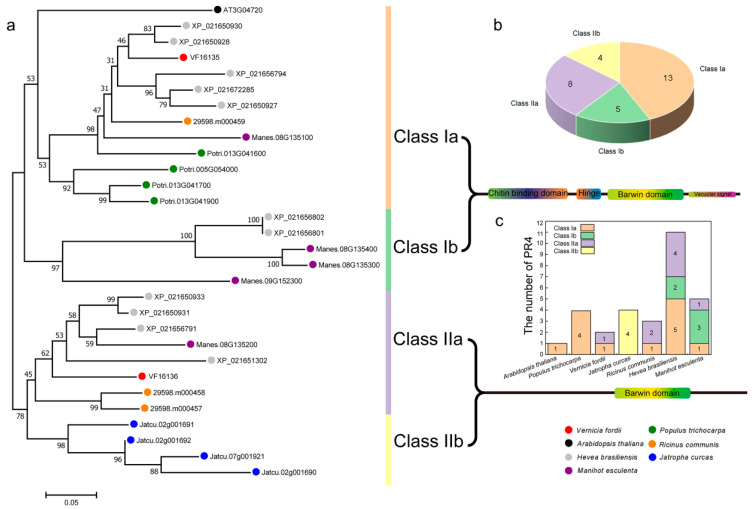
Phylogenetic relationship of *PR-4* in seven species. (**a**) Phylogenetic relationship tree. Different bootstrap values represented the credibility of the branch. Black dot: *Arabidopsis thaliana*; grey dot: *Hevea brasiliensis*; red dot: *Vernicia fordii*; orange dot: *Ricinus communis*; pink dot: *Manihot esculenta*; green dot: *Populus trichocarpa*; blue dot: *Jatropha curcas*. (**b**) Four main categories of membership statistics. Different colors pie chart indicated different classes. Orange, green, purple, and yellow pie charts represented Class Ia, Class Ib, Class Iia, and Class IIb, respectively. (**c**) Statistics *PR-4* classifications in different species.

**Figure 3 plants-12-03154-f003:**
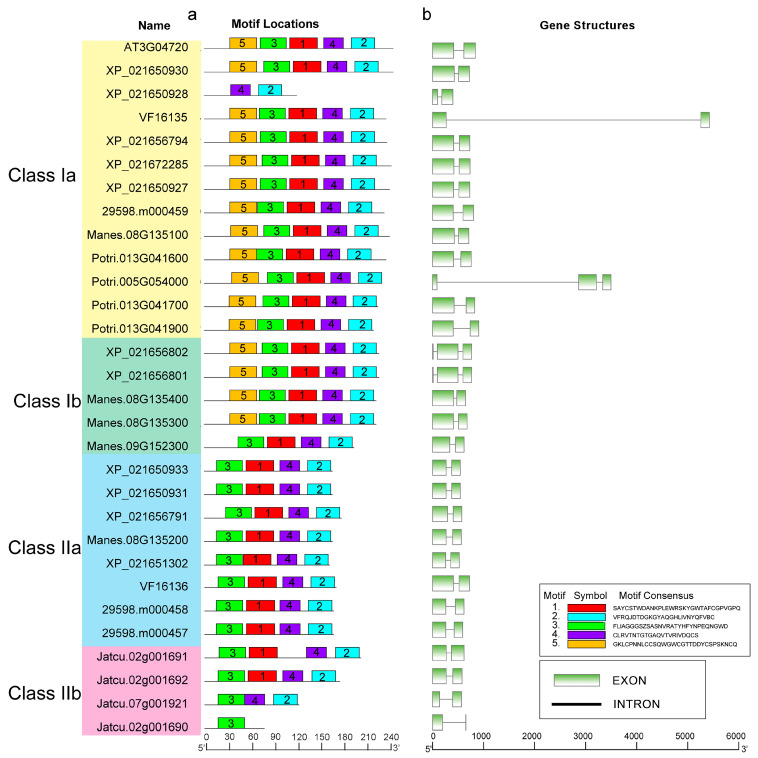
Conserved motifs and gene structures of *PR-4* gene family. (**a**) *PR-4* motif locations; 5 motifs were identified in the 30 protein sequences of *PR-4* family, each motif was shown as a box in one of 5 different colors (**b**) *PR-4* gene structures. Green boxes and black lines represented exons and introns.

**Figure 4 plants-12-03154-f004:**
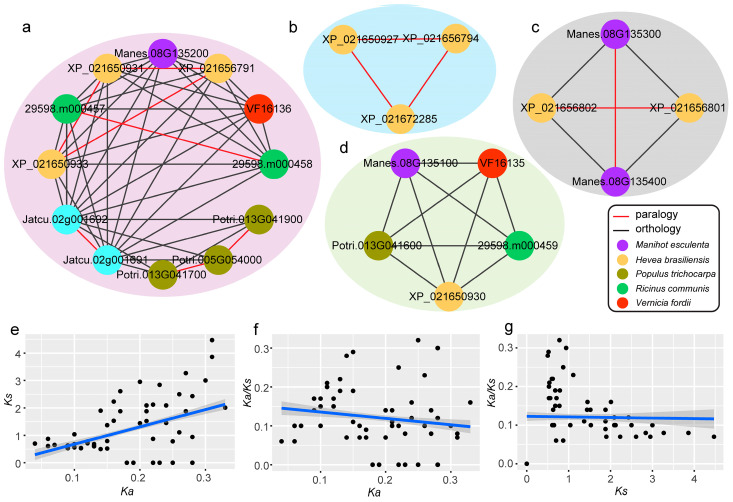
Analyses of paralogous and orthologous *PR-4* genes and Ka, Ks, and Ka/Ks. (**a**–**d**) Analyses of paralogous and orthologous *PR-4* genes, circles of different colors represented different species. (**e**–**g**) Correlative relation analyses of Ka, Ks, and Ka/Ks. (Ka/Ks) > 1 was indicated positive selection, Ka/Ks = 1 was indicated neutral selection, and Ka/Ks < 1 was indicated purifying selection.

**Figure 5 plants-12-03154-f005:**
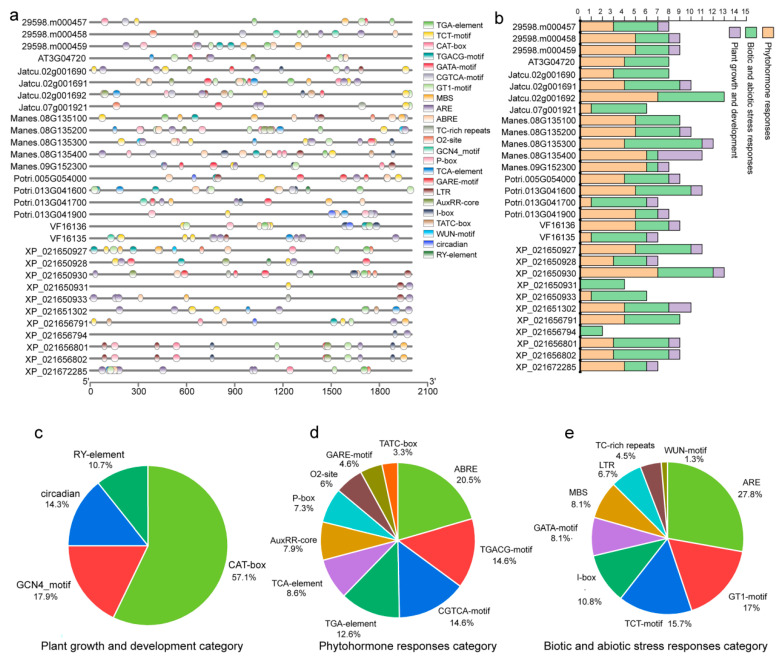
The cis-acting element analysis of *PR-4* family genes. (**a**) The different cis-acting element location analysis. (**b**) The different colored histograms represented the sum of the cis-acting elements in each category. (**c**–**e**) Pie charts of different sizes indicated the ratio of each promoter element in each category, respectively.

**Figure 6 plants-12-03154-f006:**
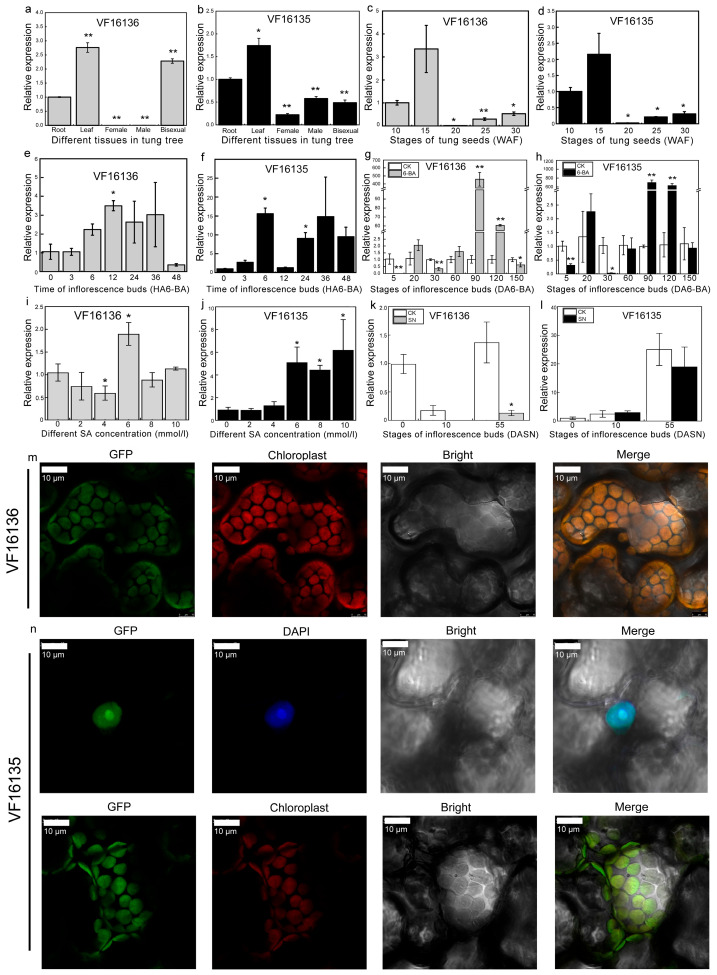
Expression pattern of *PR-4* genes in tung tree. (**a**–**d**) The relative expression of genes in different tissues and different stages of tung seeds. (**e**–**h**) The relative expression of genes in different times and different developmental stages after 6-BA treatment of inflorescence buds. (**i**,**j**) The relative expression of genes in inflorescence buds treated with SA at different concentrations. (**k**,**l**) The relative expression of genes in inflorescence buds treated with SN at different developmental stages of inflorescence buds. The data were representative of three independent biological replicates, and all data points indicated the mean ± standard error (SE) of the three biological repeats. Significant difference: *, *p* < 0.05; **, *p* < 0.01. (**m**) Subcellular localization analysis of *VF16136* in tobacco leaves. The green light represented the GFP carried by *VF16136*, the red light represented the chloroplast autofluorescence, and the merge represented the co-localization of GFP and chloroplast autofluorescence in bright. (**n**) Subcellular localization analysis of *VF16135* in tobacco leaves. The green light represented the GFP carried by *VF16135*, the 4′,6-diamidino-2-phenylindole (DAPI) was used as the nuclear dye. The length of the scale bar is 10 μm.

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
