# Peer review of "Phylogenetic Analysis of the PR-4 Gene Family in Euphorbiaceae and Its Expression Profiles in Tung Tree (Vernicia fordii)"

_plants, 2023, doi:10.3390/plants12173154_

Round 1

Reviewer 1 Report

1. The rationale for the work carried out in V. fordii is not clear to me. The authors state that they are interested in the functions of the PR4 proteins in development but no information is provided on previous work showing that this is important in other species-all examples are in relation to pathogenesis. Therefore why would this be important in V. fordii?

More background on PR4 proteins in general is lacking for example previous phylogeny analysis for example: Maia LBL et.al, (2021) Structural and Evolutionary Analyses of PR-4 SUGARWINs Points to a Different Pattern of Protein Function. Front. Plant Sci. 12:734248. doi:10.3389/fpls.2021.734248

Information from previous reports on the localization of PR4 proteins is also lacking.

2. More information of the importance of the Euphorbiaceae and in particular tung tree is missing in the introduction. There are no clear criteria as to why the species analyzed for PR4 proteins were chosen in terms of genetic relationships.

3. Any relationship between PR4 and developmental processes are very speculative with no basis in the experimental results-only the effects of 6-BA which could even be a stress response. This comment is also supported by figure 5 where the majority of motifs are related to stress and phytochrome responses and not to plant growth and development.

4. Figure 6 must be greatly improved: section i could be removed, the micrographs should be repeated-it is impossible to determine what is shown in the photographs. They are not typical of similar experiments carried out in N. benthamiana, the cells and organelles cannot be properly distinguished and nuclear staining is incongruent. In general the images must be greatly improved. The figure legend does not describe the figure adequately.

5. The discussion section must also be greatly improved to comment on the most significant points: for example why some species need different forms of PR4 and others not. The chloroplast and nuclear localization of the proteins-is this expected, is this consistent with previous reports and with the proposed functions of the proteins. Why do some PR4s have vacuolar localization signals etc. etc.

Small grammatical mistakes are present throughout the manuscript and should be addressed to facilitate reading. Also at least in one case statements seem to be contradictory leading to confusion for the reader. For example lines 80-81 state "a signal peptide consisting of 23 to 26 amino acids in the C-terminus." However this is confusing since although the vacuolar signal is at the C-terminus, the signal peptide is at the N-terminus as far as I can tell.

Reviewer 2 Report

The work by Yang et. al studies structure and phylogenetics evolution of Pathogenesis-related protein-4 (PR-4)  including domain organization, gene duplications and  cis-acting elements in seven plant species, specially focused in euphorbicaeae family and in particular tung tree (Vernicia fordii). The study includes also expression studies in different tissues, seed development stages, and inflorescence buds treated with 6-BA of two genes of V. fordii. The results contribute to the state of the art of genes structure of this gene family, especially in euphorbiaceae family, complementing previous functional studies conducted by  the authors. Although some aspects should be attended to improve the manuscript.

_  Title:  can be modified in order to better adjust to results presented “Phylogenetic analysis…” intead of “Characteristic analysis…”

-Introduction sections should be ordered and completed to highlight the phylogenetic analysis in euphorbiaceae.  Is is necessary to refer and justify the species selected for the study, those that belong to the euphorbiaciaea and those that were selected as reference species. It is necessary to clarified why Populus was included in the analysis. Also if the work focusses in the roll of PR-4 in plant development and not in pathogenesis responses as authors affirmed, citations of previous works should be consistent to this aim, instead of listing studies with references to pathogen responses.

-Figure legends should be completed to be self-explicative. Especially Figure 6 requires references to treatments performed and presented in graphs

-Taking into consideration that expression studies are focused in the roll of PR-4 genes on development it will be interesting to include gene expression studies in response to other hormonal treatments as MeJA or SA to complement the studies conducted with 6-BA. These studies arise as crucial attending to results of expression studies in other species and also to complement the studies on promoter motives described in this work.

 English language should be improved in all sections

Round 2

Reviewer 1 Report

Previous comments have been taken into account and the text and figures have been modified adequately as suggested. 

Some very minor corrections needed.

Reviewer 2 Report

Authors has attended most of the recommendations and suggestion pointed out in the first reviewed. Additional experiments, including SA treatments, were also incorporated and discussed

Attending to these comments, the manuscripts in the new version is suitable for publication

The English style was considerable improved

There is only one remaining observation to correct:

In Line 174, delete the word “was" : Cis-regulatory elements was played crucial roles in regulating plant development